# Point-of-care C-reactive protein measurement by community health workers safely reduces antimicrobial use among children with respiratory illness in rural Uganda: A stepped wedge cluster randomized trial

Emily J. Ciccone[1]*, Di Hu[2], John S. Preisser[2], Caitlin A. Cassidy[3], Lydiah Kabugho[4], Baguma Emmanuel[4], Georget Kibaba[4], Fred Mwebembezi[4], Jonathan J. Juliano[1,3], Edgar M. Mulogo[4☯], Ross M. Boyce[1,3,4☯]

1 Division of Infectious Diseases, University of North Carolina School of Medicine, Chapel Hill, North Carolina, United States of America, 2 Department of Biostatistics, University of North Carolina Gillings School of Global Public Health, Chapel Hill, North Carolina, United States of America, 3 Department of Epidemiology, University of North Carolina Gillings School of Global Public Health, Chapel Hill, North Carolina, United States of America, 4 Department of Community Health, Mbarara University of Science and Technology, Mbarara, Uganda

☯ These authors contributed equally to this work.
* emily_ciccone@med.unc.edu

## Abstract

### Background

Acute respiratory illness (ARI) is one of the most common reasons children receive antibiotic treatment. Measurement of C-reaction protein (CRP) has been shown to reduce unnecessary antibiotic use among children with ARI in a range of clinical settings. In many resource-constrained contexts, patients seek care outside the formal health sector, often from lay community health workers (CHW). This study's objective was to determine the impact of CRP measurement on antibiotic use among children presenting with febrile ARI to CHW in Uganda.

### Methods and findings

We conducted a cross-sectional, stepped wedge cluster randomized trial in 15 villages in Bugoye subcounty comparing a clinical algorithm that included CRP measurement by CHW to guide antibiotic treatment (STAR Sick Child Job Aid [SCJA]; intervention condition) with the Integrated Community Care Management (iCCM) SCJA currently in use by CHW in the region (control condition). Villages were stratified into 3 strata by altitude, distance to the clinic, and size; in each stratum, the 5 villages were randomly assigned to one of 5 treatment sequences. Children aged 2 months to 5 years presenting to CHW with fever and cough were eligible. CHW conducted follow-up assessments 7 days after the initial visit. Our primary outcome was the proportion of children who were given or prescribed an antibiotic at the initial visit. Our secondary outcomes were (1) persistent fever on day 7; (2) development

**Data Availability Statement:** Data from this study are jointly owned by the University of North Carolina (UNC) and Mbarara University of Science and Technology (MUST). Data cannot be shared publicly because institutional research ethics boards require individual permission for data sharing. Deidentified participant data that support the results are available provided the investigator proposes a methodologically sound analysis of the data, has approval from an Institutional Review Board (IRB), Independent Ethics Committee (IEC), or Research Ethics Board (REB), as applicable, and executes a data use/sharing agreement with UNC and MUST. Researchers may apply for data access by contacting the UNC IRB at irb_questions@unc.edu.

**Funding:** This study was funded by an Early Career Award from the Thrasher Research Fund (#15206; https://www.thrasherresearch.org/), a COVID-19 Supplement Award from the UNC School of Medicine Office of Research, and the Herman and Louise Smith Professorship in Medicine fund of the UNC Infectious Diseases Division Chair, all to EJC. In addition, EJC was supported by the National Heart, Lung, and Blood Institute through Grant Award Number [5T32HL007106] during the study period and is currently supported by the National Institute of Allergy and Infectious Diseases through Grant Award Number [1K23AI173658]. RMB is currently supported by the National Institute of Allergy and Infectious Diseases through Grant Award Number [5K23AI141764], and JJJ is currently supported by the National Institute of Allergy and Infectious Diseases through Grant Award Number [5K24AI134990]. The work of DH and JP was supported by the National Center for Advancing Translational Sciences (NCATS), National Institutes of Health, through Grant Award Number [UM1TR004406]. The sponsors and funders did not play any role in the study design, data collection and analysis, decision to publish, or preparation of the manuscript.

**Competing interests:** The authors have declared that no competing interests exist.

**Abbreviations:** AMR, antimicrobial resistance; ARI, acute respiratory illness; bpm, breaths per minute; CHW, community health worker; CRP, C-reactive protein; GEE, generalized estimating equation; HR, heart rate; iCCM, Integrated Community Care Management; LMIC, low- and middle-income country; MAEE, matrix-adjusted estimating equation; mRDT, malaria rapid diagnostic test; MUAC, mid-upper arm circumference; RR, respiratory rate; SCJA, Sick Child Job Aid; WHO, World Health Organization.

of prespecified danger signs; (3) unexpected visits to the CHW; (4) hospitalizations; (5) deaths; (6) lack of perceived improvement per the child's caregiver on day 7; and (7) clinical failure, a composite outcome of persistence of fever on day 7, development of danger signs, hospitalization, or death.

The 65 participating CHW enrolled 1,280 children, 1,220 (95.3%) of whom had sufficient data. Approximately 48% (587/1,220) and 52% (633/1,220) were enrolled during control (iCCM SCJA) and intervention periods (STAR SCJA), respectively. The observed percentage of children who were given or prescribed antibiotics at the initial visit was 91.8% (539/587) in the control periods as compared to 70.8% (448/633) during the intervention periods (adjusted prevalence difference −24.6%, 95% CI: −36.1%, −13.1%). The odds of antibiotic prescription by the CHW were over 80% lower in the intervention as compared to the control periods (OR 0.18, 95% CI: 0.06, 0.49). The frequency of clinical failure (iCCM SCJA 3.9% (23/585) v. STAR SCJA 1.8% (11/630); OR 0.41, 95% CI: 0.09, 1.83) and lack of perceived improvement by the caregiver (iCCM SCJA 2.1% (12/584) v. STAR SCJA 3.5% (22/627); OR 1.49, 95% CI: 0.37, 6.52) was similar. There were no unexpected visits or deaths in either group within the follow-up period.

## Conclusions

Incorporating CRP measurement into iCCM algorithms for evaluation of children with febrile ARI by CHW in rural Uganda decreased antibiotic use. There is evidence that this decrease was not associated with worse clinical outcomes, although the number of adverse events was low. These findings support expanded access to simple, point-of-care diagnostics to improve antibiotic stewardship in rural, resource-constrained settings where individuals with limited medical training provide a substantial proportion of care.

## Trial registration

ClinicalTrials.gov NCT05294510. The study was reviewed and approved by the University of North Carolina Institutional Review Board (#18–2803), Mbarara University of Science and Technology Research Ethics Committee (14/03-19), and Uganda National Council on Science and Technology (HS 2631).

## Author summary

### Why was this study done?

- Globally, antibiotics are frequently used to treat children with fever and respiratory symptoms. However, most of these illnesses are likely viral and/or self-resolving and therefore do not require antibiotics. This overuse can lead to the development and spread of antimicrobial resistance (AMR).

- Measurement of C-reactive protein (CRP) levels in blood can decrease antibiotic use in children with fever and respiratory illness in health care facilities in low- and middle-income countries (LMICs).

- Many children in sub-Saharan Africa, when ill, present first to lay community health workers, outside of the formal health care system.

## What did the researchers do and find?

- We conducted a stepped wedge cluster randomized trial involving 65 community health workers (CHW) in 15 villages in rural western Uganda that compared the Integrated Community Care Management (iCCM) algorithm currently in use by CHW in the region with an amended study algorithm (STAR Sick Child Job Aid [SCJA]) that included semiquantitative CRP measurement by CHW to guide antibiotic treatment decisions among children presenting with fever and respiratory symptoms. We enrolled 1,280 children, 1,220 of whom were included in the analyses. Our primary outcome of interest was antibiotics given or prescribed at the initial visit. As secondary outcomes, we assessed for adverse effects at 7 days after initial visit to the CHW, including clinical failure (a composite outcome that included persistent fever, development of danger signs, hospitalization, or death) and perceived improvement by the child's caregiver.

- We observed a decrease in antibiotic use of 24.6% (95% CI: −36.1%, −13.1%) in the SCJA (intervention) group compared to the iCCM (control) group.

- There was no statistical difference in rates of clinical failure, perceived improvement by the caregiver, or other safety outcomes at day 7, although the number of these events in both groups was small, and therefore, our estimates are imprecise.

## What do these findings mean?

- This study shows that point-of-care CRP testing by CHW with limited clinical training outside of the formal healthcare system is feasible and has the potential to decrease unnecessary antibiotic use. We did not observe a concomitant increased risk of adverse effects in the SCJA group, although our study was not specifically designed to detect a difference in these outcomes.

- The results provide rationale for larger-scale studies of clinical algorithms for CHW including CRP that focus on safety outcomes, implementation, and cost-effectiveness.

- This research supports the use of simple, point-of-care diagnostics in rural, resource-constrained settings to improve antibiotic stewardship.

## Introduction

The global spread of antimicrobial resistance (AMR) represents a growing threat to child health [1,2]. A primary driver of AMR is the inappropriate use of antibiotics [3], which are frequently prescribed to treat pediatric febrile acute respiratory illness (ARI) in resource-constrained settings [4–9]. A large proportion of these episodes are likely caused by self-limited and/or viral infections, and therefore, unlikely to need antibiotic treatment [10,11]. Yet at the

same time, timely and accurate identification of the children at high risk for bacterial infection is critical to minimizing morbidity and mortality from pneumonia. Differentiating viral respiratory infections from bacterial pneumonia in resource-constrained contexts is challenging, as robust laboratory and diagnostic imaging infrastructure is not often present.

In low- and middle-income countries (LMICs) like Uganda, lay community health workers (CHW) are assuming a greater role in the initial care of children with the goal of improving healthcare access to basic services. Many of these individuals are trained to evaluate and treat children under Integrated Community Case Management (iCCM) programs, a care strategy developed by the World Health Organization (WHO) to reduce mortality in young children related to malaria, pneumonia, and diarrhea [12–14]. Evaluations of such programs have demonstrated that CHW are able to provide a high quality of care for malaria by utilizing malaria rapid diagnostic tests (mRDTs) for diagnosis [15–17]. The quality of pneumonia management is less consistent [15,18]. According to the iCCM algorithm, children who have both cough and fast breathing (i.e., tachypnea) are diagnosed with pneumonia and treated with antibiotics. However, it is difficult to accurately measure the respiratory rate, and its assessment is influenced by contextual factors [19,20]. Furthermore, tachypnea can occur with both bacterial and viral infections. This dependence on syndromic diagnosis contributes to both (1) an overuse of antibiotics, which can drive AMR; and (2) an under-recognition of children who are at high-risk for bacterial infection [20,21].

One tool for more accurately targeting antibiotic therapy is the measurement of the clinical biomarker, C-reactive protein (CRP) [22–28]. CRP is released in response to inflammation and to a greater extent in bacterial as compared to viral infection [29]. Its performance characteristics for ruling bacterial infection in or out vary by threshold chosen; a cut-off of approximately 40 mg/l has been associated with a high negative predictive for radiologic end-point pneumonia [23]. In LMIC specifically, among febrile children with ARI, clinical decision algorithms that included CRP implemented by clinicians in outpatient clinics have substantially reduced antibiotic use without increasing adverse outcomes [30–32]. In addition, employing a cut-off of 40 mg/l for determining whether antibiotics are administered in the context of febrile illness has been shown to be safe [33]. However, other studies have noted a much more modest impact on antibiotic use [27,33]. Furthermore, the reliance of most CRP assays on an analyzer requiring electricity and the cost of the test have limited their implementation outside of more established laboratory facilities or resource-rich contexts.

Therefore, while effective in a clinic setting, more data on the utility of CRP to optimize the management of pediatric ARI in resource-constrained settings are needed. To date, CRP measurement to inform antibiotic decision-making in the context of iCCM care provided by lay CHW, outside of the formal health sector, has not been evaluated, even as CHW assume a greater role in caring for children with ARI. To address these knowledge gaps, we conducted a stepped wedge cluster randomized trial of an adapted iCCM algorithm that included point-of-care, lateral flow-based CRP measurement by CHW evaluating children with febrile ARI in rural western Uganda. We chose this study design to (1) facilitate cluster recruitment and enhanced acceptability of the study among the Village Health Teams; (2) pragmatically assess effectiveness of the intervention; and (3) conduct a randomized evaluation of the intervention (as opposed to a weaker, observational, pre-post study) with relatively limited funding [34,35].

## Methods

### Study setting

We conducted this study in the Bugoye subcounty of Kasese District in western Uganda (**Fig 1**). The region has highly variable geography; in the western villages, there are deep river

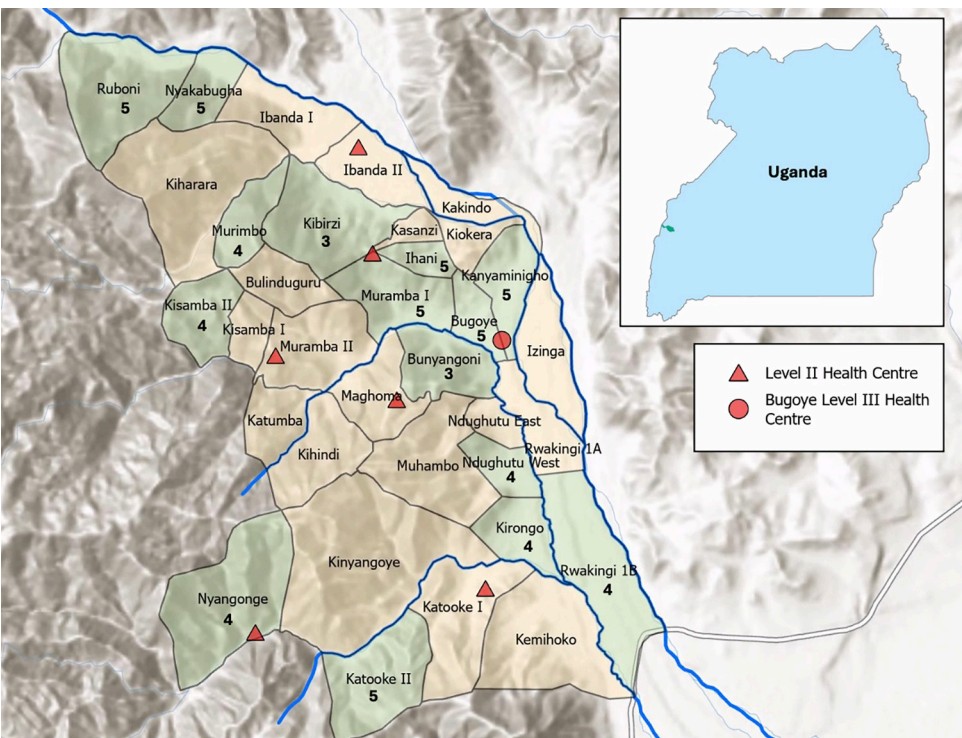

**Fig 1. Map of Bugoye subcounty in Kasese District in western Uganda.** All 34 villages within Bugoye subcounty are shown. Villages that participated in the study are shown in green with the number of CHW who participated from that village. Locations of Level II and Level III health facilities indicated by red triangles and a circle, respectively. Major rivers are shown in blue. The location of Bugoye subcounty within Uganda is shown in the inset. Map was generated using ArcGIS with publicly available basemap from Esri (https://www.arcgis.com/home/item.html?id=a52ab98763904006aa382d90e906fdd5). CHW, community health worker.

valleys and steep hills of up to 2,000 meters in elevation, whereas the terrain in the eastern villages is level and at lower elevations. A quarter of the subcounty's population of approximately 35,000 residents are children under the age of 5 years [36].

In recent years, the Government of Uganda has implemented an iCCM program consisting of teams of volunteer CHW, known as Village Health Teams. The government program began operating in Bugoye subcounty in 2013, and at the time this study was conducted in 2021 to 2022, external groups were also supporting additional CHW in 22 of the subcounty's 34 villages. All CHW participate in a 5-day, intensive training per Ugandan Ministry of Health guidelines on receiving patients and caregivers, assessing children under age 5, managing and treating diarrhea, pneumonia, and malaria, using mRDTs, referring cases that CHW are not trained to manage, recording data, and patient follow-up. They also undergo quarterly, half-day refresher trainings. ARI, specifically defined as cough or fast breathing, is common in Bugoye subcounty—it was the second most frequent reason for a CHW visit, representing 45% of the 18,430 visits documented between April 2014 and December 2018 [37]. The routine immunization schedule in Uganda includes 3 doses each of *Haemophilus influenzae* type B (Hib) and 10-valent pneumococcal conjugate vaccine (PCV-10) [38]. In the region that includes Bugoye subcounty, full vaccine coverage is estimated at 75% for Hib and 54% for PCV-10 [39]. The prevalence of HIV among children age 0 to 14 in Uganda is estimated at 0.5% (95% CI: 0.3 to 0.8) [40].

## Study design

We performed a stepped wedge, cluster randomized trial called the Stewardship for Acute Respiratory Illness or STAR study. Villages (clusters) were switched monthly from control to intervention in a staggered fashion. Each treatment sequence included 3 villages, and there were 5 sequences (15 villages total). The study had 6 observation periods of approximately 1 month each, with 3 villages randomly assigned to cross over from control to intervention at each step (Fig 2). We employed a cross-sectional design where different eligible children were enrolled during each month. The cross-sectional aspect meant that while individual children were in the trial for a short time (7 days of follow-up), new children were continually enrolled as they presented to the CHW. The CONSORT checklist for stepped wedge trials is included as S1 Checklist [41].

## Randomization

Considering distance from Bugoye Health Centre III (red circle in Fig 1), altitude, and volume of patients seen by the CHW each month, the study team selected 15 of the 22 villages with

| Sequence | Village | Period (Number of Days) | | | | | | Total |
| | | 1 (30) | 2 (40) | 3 (27) | 4 (27) | 5 (28) | 6 (31) | |
|---|---|---|---|---|---|---|---|---|
| 1 | Buny | 23 | 30 | 20 | 26 | 5 | 8 | 112 |
| 1 | Kib | 11 | 12 | 16 | 22 | 4 | 3 | 68 |
| 1 | Ndu | 16 | 45 | 31 | 29 | 7 | 9 | 137 |
| 2 | Mur | 15 | 12 | 11 | 17 | 8 | 8 | 71 |
| 2 | Nyak | 10 | 15 | 21 | 22 | 9 | 9 | 86 |
| 2 | Rub | 11 | 12 | 3 | 23 | 7 | 11 | 67 |
| 3 | Kan | 17 | 20 | 7 | 10 | 11 | 10 | 75 |
| 3 | Kiro | 28 | 31 | 7 | 18 | 12 | 8 | 104 |
| 3 | Nya | 9 | 15 | 17 | 27 | 8 | 6 | 82 |
| 4 | Ihan | 21 | 34 | 15 | 18 | 19 | 6 | 113 |
| 4 | Kat | 10 | 16 | 24 | 7 | 10 | 8 | 75 |
| 4 | Mir | 13 | 17 | 13 | 8 | 15 | 11 | 77 |
| 5 | Bug | 23 | 13 | 18 | 11 | 5 | 13 | 83 |
| 5 | Kis | 7 | 5 | 7 | 3 | 5 | 14 | 41 |
| 5 | Rwa | 8 | 6 | 1 | 2 | 1 | 11 | 29 |
| Total | | 222 | 283 | 211 | 243 | 126 | 135 | 1220 |

**Fig 2. Design of the STAR stepped wedge cluster randomized trial and enrollment by village and period.** The study began enrollment on November 1, 2021 and ended enrollment on May 12, 2022 with each period lasting approximately 1 month. The number of days in each period is in parentheses below the period number (please see S2 Fig for exact period switch dates). Yellow blocks represent control periods; blue blocks are intervention periods. The number shown within the blocks is the total number of participants enrolled for each village and period.

additional external support for inclusion in the trial. Each of the 15 villages selected had a robust and well-established CHW program with 3 to 5 CHWs per village.

To ensure balance of factors that may influence ARI epidemiology and malaria transmission intensity, we performed stratified randomization by altitude, distance from the subcounty's regional health facility (Bugoye), and village size at the level of the village across treatment sequences of conditions (control v. intervention). The estimated number of eligible children seen per year by the CHW in each village was used as a proxy for village size. Emphasizing the first 2 factors while allowing some overlap in size, we first created 3 strata of 5 villages each: (1) low altitude, proximal, "large" villages (>115 eligible children seen per year); (2) low altitude, mid-distance, "medium size" villages (90 to 140 eligible children); and (3) high altitude, distal, "small" villages (<120 eligible children seen). Into each of the 5 treatment sequences, we randomly selected 1 village from each stratum. The specifics of the villages included in each stratum are shown in **S1 Table**.

## Study intervention

During the control periods (yellow blocks in **Fig 2**), the routine iCCM algorithm, herein referred to the iCCM Sick Child Job Aid (SCJA), was used for evaluation of enrolled children presenting with cough and fever. Per the iCCM SCJA, the CHW assessed all children for danger signs defined as severe chest in-drawing, inability to breastfeed or drink, and/or decreased level of consciousness, and if present, referred them to the nearest health facility after providing pre-referral management including a first dose of antibiotics. If no danger signs were present, a respiratory rate was measured and, if elevated for age, antibiotic treatment for bacterial pneumonia (5 days of dispersible amoxicillin tablets) should be given (or prescribed, in the case of stockouts). The full routine iCCM algorithm used by the Uganda Ministry of Health and the CHW program in Bugoye subcounty is described elsewhere and available on the Ministry of Health website [12,42–44].

In the intervention periods (blue blocks in **Fig 2**), the CHW used a modified SCJA (herein referred to as the STAR SCJA) that included CRP testing to guide antibiotic treatment decisions (see **S1 Fig** for full algorithm). In brief, the CHW assessed all enrolled children for danger signs and performed an mRDT and a similar, lateral flow-based, point-of-care, semiquantitative CRP test (Actim CRP, Medix Biochemica) per manufacturer instructions. If danger signs were present, the patient was referred to the nearest health facility and received pre-referral management, including antibiotics, per iCCM protocols regardless of CRP result. If no danger signs were noted, the CHW determined if antibiotic treatment for pneumonia was to be administered based on the CRP result. If CRP ≥40 mg/l, they dispensed 5-day course of amoxicillin per local guidelines. If CRP <40 mg/l, the CHW advised symptomatic care alone including paracetamol for fever and additional fluids to maintain hydration.

Prior to study start, all participating CHW underwent a hands-on training on the performance and interpretation of the Actim CRP test by laboratory personnel from Bugoye Health Centre III as well as the study staff and principal investigator, all of whom had experience using the Actim CRP test. The CHW were also given written instructions including pictures to refer to if needed during study implementation and participated in refresher trainings between each period.

In both control and intervention periods, CHW measured mid-upper arm circumference (MUAC) for each child to assess nutritional status. They advised all caregivers to bring the child to a health facility if, after the initial evaluation, they could not feed or drink, had a fever for more than a week, or developed chest in-drawing. They also conducted an in-person or phone follow-up assessment at 7 days with all enrolled children to assess for secondary outcomes (see below).

## Study participants

All the CHW living in the 15 villages were approached regarding participation as study research assistants. A total of 67 CHWs underwent the initial training; 2 elected not to participate prior to the start of enrollment. Therefore, all children were enrolled by 65 CHW (**Fig 1**).

Children were eligible for the study if they were aged 2 months to 5 years, evaluated by one of the study CHW in one of the study villages, and had an acute respiratory illness defined as fever (documented (temperature >38°C) or subjective fever in the last 7 days) AND fast breathing (respiratory rate >30 breaths per minute (bpm)) OR cough. Children were excluded if a guardian was not present to provide consent. Those individuals opting not to participate underwent evaluation and treatment per iCCM guidelines (iCCM SCJA) regardless of whether the village was in a control or intervention period. Children could participate in the study more than once if the previous illness had completely resolved, and they were re-presenting for a new episode of ARI. Determination of resolution of symptoms was at the discretion of the CHW.

## Clinical definitions

Fast breathing was a subjective assessment reported by the caregiver or the participant. Respiratory rate (RR) was determined via a manual breath count by the CHW using the timer provided by the CHW program. Tachypnea for age was defined as an RR of >50 bpm for children aged 2 to 11 months and >40 bpm for children aged 1 to 5 years per local standards of care [12,45]. The study provided a thermometer and MUAC tape to each CHW; one CHW from each village was provided with a pulse oximeter to measure heart rate (HR) and oxygen saturation (SpO2). Tachycardia was defined as a HR of >160 beats per minute for age 2 months to 1 year, >150 for ages 1 to 3 years, and >140 for ages 3 to 5 years per the WHO [46]. Hypoxia was defined as an SpO2 of less than 90%. Severe acute malnutrition and moderate acute malnutrition were defined as an MUAC of <115 mm and ≥115 mm to <125 mm, respectively, per WHO standards [47].

## Outcomes

The primary child-level outcome was whether an antibiotic was given or prescribed by the CHW during the initial assessment of the child. In addition, we performed a post hoc subgroup analysis of our primary outcome by mRDT result status. Finally, we conducted an ancillary analysis of whether an antibiotic was given or prescribed either at the initial (baseline) visit or during the follow-up period (which was a prespecified secondary outcome).

There were 7 additional prespecified secondary outcomes: (1) persistent fever on day 7; (2) development of danger signs; (3) unexpected visits to the CHW during the 7-day follow-up period; (4) hospitalizations; (5) deaths; (6) lack of perceived improvement per the child's caregiver on day 7; and (7) clinical failure, which was a composite outcome of persistence of fever at the follow-up assessment, development of danger signs, hospitalization, or death. Post hoc secondary outcomes included (1) an alternate clinical failure composite outcome of lack of perceived improvement by caregiver on day 7, development of danger signs, and/or hospitalization; (2) a composite outcome of need for further outpatient evaluation during the study follow-up period (visit to a traditional healer, drug shop, or clinic for the same problem); (3) the persistence of symptoms other than fever at day 7; and (4) presence of tachypnea at day 7.

## Statistical methods

Demographic and clinical characteristics were summarized with descriptive statistics and the proportion of children who were given or prescribed antibiotics by treatment condition and

village was calculated. The analysis of the primary outcome employed a marginal logistic regression model estimated within a generalized estimating equations (GEEs) framework to describe how the probability of antibiotic use changed across the subsets of the population defined by the intervention and control cluster-periods (see **S1 Appendix** for details). The analysis performed with the SAS macro GEEMAEE [48] treated villages as clusters; fitted a logistic model that included period and strata as categorical variables and an indicator variable for intervention; specified a within-village, nested exchangeable correlation structure having one within-period and one between-period correlation parameter to account for correlation decay over time; and made two kinds of small-sample bias-corrections considering that 15 clusters (villages) is too small a number to otherwise reliably employ standard GEE. Thus, the analysis used bias-corrected variance estimators for model parameter estimates, and bias corrections to the estimated within-cluster correlations via a procedure known as matrix-adjusted estimating equations (MAEEs) [49,50]. From the logistic model, the odds ratio for the intervention effect and its 95% confidence interval were computed (**S1 Appendix**). Similar models were fitted for antibiotic use in mRDT positive and negative subgroups and in ancillary analysis of any antibiotic use at baseline or follow-up; the identity link was used in models that estimated prevalence differences. There were only 3 participants enrolled during an intervention period who were managed as in the control condition (using the iCCM SCJA). As this number was so small, these individuals were included as-treated. Analyses were performed using SAS 9.4 and R 4.3.1.

For secondary outcomes, which were uncommon events in that they occurred in less than 10% of study participants, we used a stratified logistic regression model with linear period effects, the intervention indicator variable and village as fixed effects. We used exact conditional inference with the EXACT statement in SAS Proc Logistic with village managed as a stratification factor. An exception was tachypnea; it was analyzed with GEE/MAEE like the primary outcome since it was not an uncommon outcome.

## Power calculation and sample size

Prior to initiation of the trial, statistical power was computed for the dichotomous primary outcome based on GEE analysis and the stepped wedge design assuming 15 clusters (villages) transitioning from control to intervention condition in 5 waves of 3 clusters each and 6 total periods of follow-up, as in **Fig 2**, except that an equal number of children was assumed in each sequence-period [48,49]. Assuming 12 children recruited per village-month (total sample size of 1,080), an antibiotic use rate of 80% under the control condition and 60% under the intervention condition, the study had 87% power corresponding to an odds ratio of 0.375 with two-sided $\alpha = 0.05$ GEE Wald tests assuming a zero-slope temporal trend, a within-period correlation of 0.10 and a between-period correlation of 0.05; note that the power calculation adjusted for time but, in the absence of empirical data suggesting otherwise, its trend parameter was set to zero (see **S1 Appendix**).

## Research ethics review and trial registration

The study was reviewed and approved by the University of North Carolina Institutional Review Board (#18–2803), Mbarara University of Science and Technology Research Ethics Committee (14/03-19), and Uganda National Council on Science and Technology (HS 2631). Ethics review and initial study approval by the 3 committees was finalized on March 25, 2021, and the study protocol version under which participants were enrolled was approved on May 26, 2021. Study enrollment took place between November 1, 2021 and May 12, 2022. Written informed consent for each participating child was provided by a parent or guardian. Due to an

administrative error, trial registration occurred after recruitment began (ClinicalTrials.gov NCT05294510; March 23, 2022), which could have introduced investigator bias (see Discussion below). For detailed timeline of ethics review, trial registration, and protocol approval, please see **S2 Appendix**.

## Results

### Study population

A total of 1,280 children were enrolled between November 2021 and May 2022 (**Fig 3**), a time period which encompassed both rainy and dry seasons. Sixty individuals were excluded from this analysis because of missing data collection forms. Slightly more participants were enrolled during intervention periods (*n* = 633, 52%) as compared to the control periods (*n* = 587, 48%).

### Baseline characteristics

Demographic characteristics of the enrolled children and participating CHW are shown in **Tables 1 and S2**, respectively. Of note, age, sex, and family size were similar for children enrolled during control and intervention periods. A slightly higher proportion of children lived in permanent houses, and a slightly lower proportion lived in semipermanent or temporary structures, in the control as compared to the intervention group. The distribution of the village size of the enrollees was similar in the control and intervention groups, with a slightly higher proportion of children living in small villages in the intervention group.

The clinical characteristics of the participants are described in **Table 2**. Fewer participants and their caregivers in the intervention arm reported fast breathing as a symptom, whereas more had a fever documented by the clinician during the initial assessment. The proportion of children with tachypnea for age as measured by the CHW was similar in both groups, as was the frequency with which cough was reported. Among the subset of participants for whom the study CHW measured HR and SpO2, a slightly higher proportion were tachycardic for age in the intervention arm. The distribution of nutritional status and proportion of children who were mRDT-positive was similar in both groups. Antimalarial use by mRDT result is shown in **S3 Table**.

### Antibiotic use

Our primary outcome of interest was antibiotic treatment (amoxicillin) given or prescribed by the CHW at the initial visit (**Table 3**). In the control periods, CHWs gave antibiotics to 91.8% (539/587) of children as compared to 70.8% (448/633) during the intervention periods (adjusted prevalence difference −24.6%, 95% CI: −36.1%, −13.1%). The observed proportions of antibiotic use varied slightly over time but with no apparent temporal trend (**S4 and S5 Figs**). After adjusting for month of visit and stratum, the odds of antibiotic prescription by the CHW were over 80% lower in the intervention as compared to the control periods (OR 0.18, 95% CI: 0.06 to 0.49). We observed a similar trend in both mRDT-positive and mRDT-negative children (**Table 3**). Full results of the logistic regression model for antibiotic use are given in **S4 Table**.

Among those who did not receive antibiotics at the initial visit, there were numerically more participants in the intervention group who acquired antibiotics between the initial and follow-up visits, but the absolute number of additional antibiotic prescriptions was low in both groups (5.8% (10/172) v. 2.2% (1/45)). As such, analysis results of the secondary outcome of antibiotics given or prescribed at any point (initial visit or during follow-up) among children

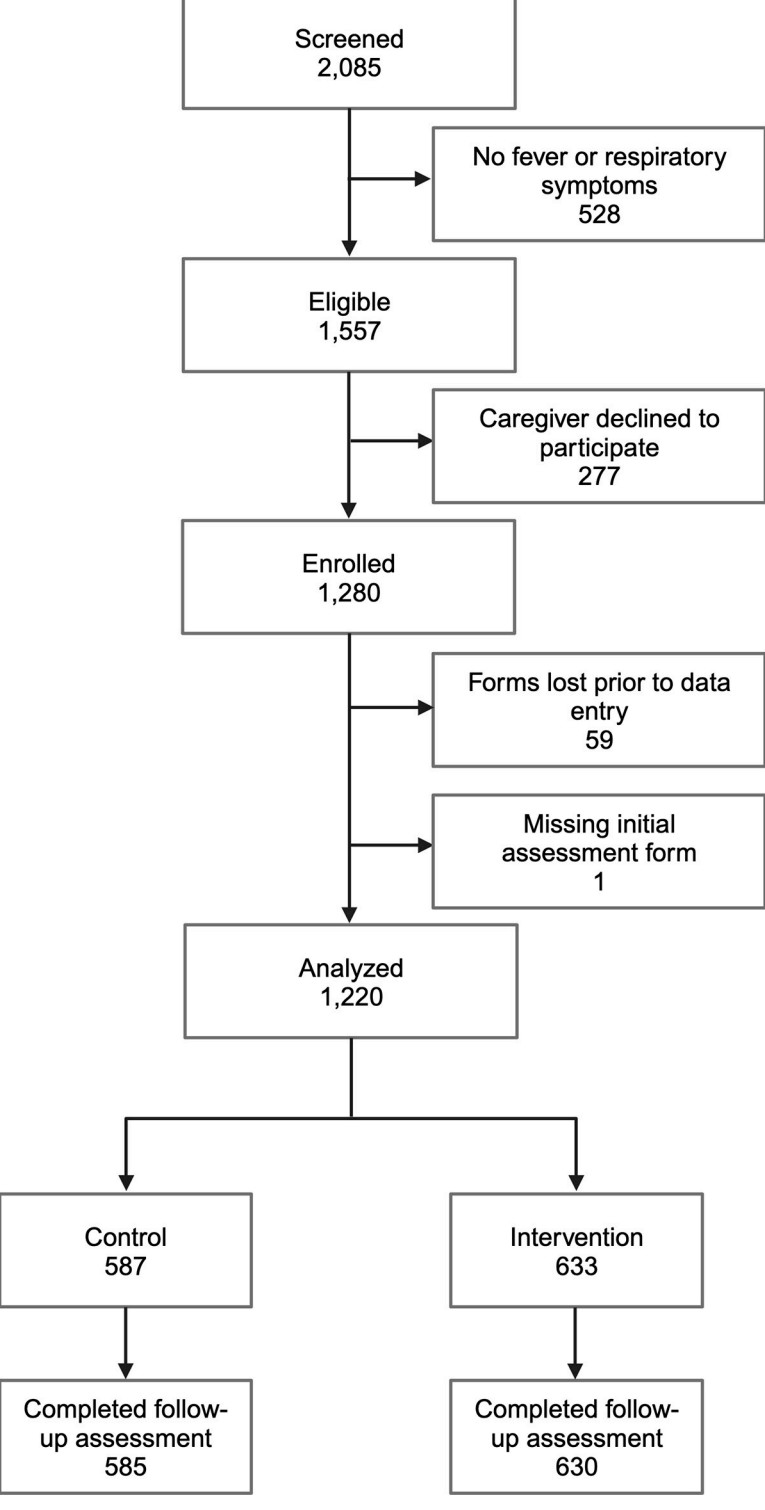

**Fig 3. Participant flow diagram.** The number of children enrolled in each period is shown by sequence (**S2 Fig**) and by village (**Fig 2**). The number of children enrolled per CHW over the course of the study ranged from 3 to 49 with a median of 16.5 (**S3 Fig**). The CHW in villages from the first treatment sequence enrolled the highest average number of participants per person (mean (SD) = 31.7 (9.2)), whereas the CHWs in villages from the last (fifth) sequence enrolled the lowest number (mean (SD) = 11.8 (7.3)). Fewer participants were enrolled in the last 2 periods than the first 4 periods (**Figs 2 and S2**). Created with BioRender.com. CHW, community health worker; SD, standard deviation.

**Table 1. Demographics of the study population.** All values shown are *n*(%) unless otherwise indicated.

| | Control (*N* = 587) | Intervention (*N* = 633) | Total (*N* = 1,220) |
|---|---|---|---|
| **Age in months** | | | |
| Median (IQR) | 24.0 (12.0, 36.0) | 24.0 (15.0, 36.0) | 24.0 (13.0, 36.0) |
| **Sex** | | | |
| Female | 310 (52.8%) | 329 (52.0%) | 639 (52.4%) |
| **Occupation of guardian** | | | |
| Subsistence farmer | 563 (95.9%) | 588 (93.0%) | 1,151 (94.4%) |
| Teacher | 10 (1.7%) | 15 (2.4%) | 25 (2.1%) |
| Businessperson | 4 (0.7%) | 9 (1.4%) | 13 (1.1%) |
| Boda boda driver | 6 (1.0%) | 4 (0.6%) | 10 (0.8%) |
| Tailor | 1 (0.2%) | 9 (1.4%) | 10 (0.8%) |
| Student | 2 (0.3%) | 1 (0.2%) | 3 (0.2%) |
| Trader | 0 (0.0%) | 3 (0.5%) | 3 (0.2%) |
| Civil servant | 0 (0.0%) | 1 (0.2%) | 1 (0.1%) |
| Engineer | 1 (0.2%) | 0 (0.0%) | 1 (0.1%) |
| LC1 chairman | 0 (0.0%) | 1 (0.2%) | 1 (0.1%) |
| Community health worker | 0 (0.0%) | 1 (0.2%) | 1 (0.1%) |
| **Family size** | | | |
| Median (IQR) | 6.0 (4.0, 8.0) | 6.0 (5.0, 8.0) | 6.0 (5.0, 8.0) |
| **Home construction** | | | |
| Permanent (bricks with iron sheets) | 278 (47.4%) | 242 (38.2%) | 520 (42.7%) |
| Semipermanent (mud with iron sheets) | 282 (48.1%) | 352 (55.6%) | 634 (52.0%) |
| Temporary (mud with grass thatched roof) | 26 (4.4%) | 39 (6.2%) | 65 (5.3%) |
| Missing | 1 | 0 | 1 |
| **Village stratum** | | | |
| 1 (large, low altitude, proximal) | 245 (41.7%) | 234 (37.0%) | 479 (39.3%) |
| 2 (medium, low altitude, mid-distance) | 177 (30.2%) | 185 (29.2%) | 362 (29.7%) |
| 3 (small, high altitude, distant) | 165 (28.1%) | 214 (33.8%) | 379 (31.1%) |

for whom data were available at both time points, were like that of the primary outcome (**Table 3**).

## Clinical failure and adverse events

Secondary outcomes assessing the safety of the STAR algorithm are detailed in **Fig 4** and **S5 Table**. Overall, we did not find evidence to suggest any relationship between the intervention and secondary outcomes as all 95% confidence intervals for odds ratios included 1.0. Specifically, clinical failure was uncommon and occurred at a slightly lower frequency in the intervention group (control 3.9% (23/585) v. intervention 1.8% (11/630); OR 0.41, 95% CI: 0.09 to 1.83). There was no statistical difference in the risk of hospitalization or persistence of fever, both of which were rare events. There were numerically fewer participants with any danger sign at the follow-up visit in the intervention arm, but this outcome was overall rare in both groups (1.7% (10/585) v. 0.5% (3/630); OR 0.32, 95% CI: 0.02 to 5.12). No unexpected visits to the CHW between initial and follow-up visits occurred, and there were no deaths during study follow-up. Few caregivers perceived that their child had not improved at the day 7 follow-up assessment (control 2.1% (12/584) v. intervention 3.5% (22/627); OR 1.49, 95% CI: 0.37 to 6.52).

**Table 2. Clinical presentation of the study participants.** All values shown are *n*(%) unless otherwise indicated.

| | Control (routine iCCM)<br>(*N* = 587) | Intervention (STAR study algorithm)<br>(*N* = 633) | Total<br>(*N* = 1,220) |
|---|---|---|---|
| **Days since first fever** | | | |
| Median (IQR) | 2.0 (1.0, 3.0) | 2.0 (2.0, 3.0) | 2.0 (1.0, 3.0) |
| **Danger signs** | | | |
| N/A—no danger signs | 562 (95.7%) | 618 (97.6%) | 1,180 (96.7%) |
| Vomiting everything | 4 (0.7%) | 6 (0.9%) | 10 (0.8%) |
| Not able to drink or breastfeed | 4 (0.7%) | 4 (0.6%) | 8 (0.7%) |
| Convulsions | 1 (0.2%) | 1 (0.2%) | 2 (0.2%) |
| Severe chest in-drawing | 4 (0.7%) | 1 (0.2%) | 5 (0.4%) |
| Very sleepy or unconscious/difficult to wake | 2 (0.3%) | 5 (0.8%) | 7 (0.6%) |
| **Symptoms** | | | |
| Skin rash | 27 (4.6%) | 36 (5.7%) | 63 (5.2%) |
| Cough | 569 (96.9%) | 627 (99.1%) | 1,196 (98.0%) |
| Headache | 116 (19.8%) | 85 (13.4%) | 201 (16.5%) |
| Runny nose | 163 (27.8%) | 184 (29.1%) | 347 (28.4%) |
| Diarrhea | 80 (13.6%) | 82 (13.0%) | 162 (13.3%) |
| Not eating | 31 (5.3%) | 43 (6.8%) | 74 (6.1%) |
| Stridor | 2 (0.3%) | 3 (0.5%) | 5 (0.4%) |
| Fever | 583 (99.3%) | 628 (99.2%) | 1,211 (99.3%) |
| Muscle aches | 2 (0.3%) | 0 (0.0%) | 2 (0.2%) |
| Fatigue | 4 (0.7%) | 0 (0.0%) | 4 (0.3%) |
| Sore throat | 2 (0.3%) | 1 (0.2%) | 3 (0.2%) |
| Joint pains | 10 (1.7%) | 7 (1.1%) | 17 (1.4%) |
| Vomiting | 41 (7.0%) | 55 (8.7%) | 96 (7.9%) |
| Fast breathing | 194 (33.0%) | 140 (22.1%) | 334 (27.4%) |
| Wheezing | 9 (1.5%) | 18 (2.8%) | 27 (2.2%) |
| Other | 1 (0.2%) | 1 (0.2%) | 2 (0.2%) |
| **Vital Signs** | | | |
| Fever[a] | 514 (87.6%) | 501 (79.1%) | 1,015 (83.2%) |
| Tachypnea[b] | 522 (89.4%) | 537 (86.1%) | 1,059 (87.7%) |
| *Missing* | 3 | 9 | 12 |
| Tachycardia[c,d] | 10 (7.2%) | 23 (19.5%) | 33 (12.8%) |
| *Missing* | 448 | 514 | 962 |
| Hypoxia[c,d] | 43 (30.7%) | 26 (22.0%) | 69 (26.7%) |
| *Missing* | 447 | 515 | 962 |
| MUAC | | | |
| *Severe acute malnutrition* | 7 (1.2%) | 10 (1.6%) | 17 (1.4%) |
| *Moderate acute malnutrition* | 8 (1.4%) | 13 (2.1%) | 21 (1.7%) |
| *Adequately nourished* | 572 (97.4%) | 608 (96.4%) | 1,080 (96.9%) |
| *Missing* | 0 | 2 | 2 |
| **Antibiotic treatment in the last 2 weeks** | | | |
| Yes | 29 (4.9%) | 47 (7.4%) | 76 (6.2%) |
| Missing | 1 | 0 | 1 |
| **Seen at another clinic for this same problem?** | | | |
| Yes | 89 (15.2%) | 139 (22.0%) | 228 (18.7%) |
| **mRDT Positive** | 310 (52.8%) | 318 (50.2%) | 628 (51.5%) |

(*Continued*)

**Table 2.** (Continued)

| | Control (routine iCCM) (N = 587) | Intervention (STAR study algorithm) (N = 633) | Total (N = 1,220) |
|---|---|---|---|
| Missing | 0 | 2 | 2 |

[a]Fever defined as an axillary temperature of >/ = 38.0˚C.

[b]Tachycardia defined as >160 bpm for age 2 months–1 year, >150 bpm for ages 1–3 years, and >140 bpm for ages 3–5 years (43).

[c]Tachypnea was defined as RR > = 50 for children 2–11 months and RR > = 40 for children 12 months or older.

[d]Hypoxia defined as oxygen saturation (SpO2) of <90%. Only 1 CHW per village was given a pulse oximeter.

bpm, breaths per minute; CHW, community health worker; iCCM, Integrated Community Care Management; mRDT, malaria rapid diagnostic test; MUAC, mid-upper arm circumference; RR, respiratory rate.

To further assess the impact and safety of the intervention, we conducted several post hoc analyses of clinical outcomes and care-seeking (**Fig 4** and **S5 Table**). Among those who had a respiratory rate measured at the follow-up visit, the proportion who were tachypneic for age was similar between control and intervention conditions (16.9% (70/414) v. 17.0% (78/460); OR 1.23, 95% CI: 0.71 to 2.15). Persistence of symptoms other than fever was rare and occurred at a similar frequency in both conditions (1.9% (11/585) v. 2.2% (14/630); OR 0.71, 95% CI: 0.14 to 3.63). Finally, there was no statistical difference in further care-seeking from another outpatient provider (traditional healer, drug shop, or outpatient clinic) for the same illness (control 5.5% (32/585) v. intervention 5.2% (33/630); OR 1.50, 95% CI: 0.54 to 4.26).

### CRP test results

The distribution of CRP testing results in the intervention group is shown in **Fig 5**. The majority of participants had CRP levels >40 mg/l (70.3%, 444/632), with the most common test

**Table 3. Comparison of antibiotic use among children under the control and intervention treatment conditions during the STAR stepped wedge cluster randomized trial (as treated analysis).** Children seen during control periods were evaluated and managed using the iCCM SCJA; children seen during the intervention condition were evaluated and managed using the STAR SCJA.

| | | Observed proportions[1] | | Adjusted intervention effect[2] (95% CI) | | | |
|---|---|---|---|---|---|---|---|
| | | Control | Intervention | Prevalence difference | Odds ratio | Estimate (SD-BC2)[3] | |
| | | | | | | Within-period | Between-period |
| Antibiotics at initial visit | All children | 539/587 (91.8%) | 448/633 (70.8%) | −24.6% (−36.1%, −13.1%) | 0.18 (0.06, 0.49) | 0.045 (0.024) | 0.040 (0.018) |
| | mRDT+ | 285/310 (91.9%) | 228/318 (71.7%) | −21.1% (−37.7%, −4.6%) | 0.18 (0.06, 0.59) | 0.063 (0.045) | 0.041 (0.032) |
| | mRDT − | 254/277 (91.7%) | 218/313 (69.6%) | −26.6% (−39.8%, −13.4%) | 0.26 (0.11, 0.61) | 0.056 (0.018) | 0.054 (0.027) |
| Antibiotics at any time | | 540/584 (92.5%) | 458/620 (73.9%) | −22.4% (−33.5%, −11.3%) | 0.17 (0.06, 0.51) | 0.049 (0.029) | 0.048 (0.023) |

[1]Three children seen during an intervention period were evaluated using the iCCM SCJA; they were included in the control condition.

[2]The prevalence difference and odds ratio estimates are based on GEE with an identity link and logit link, respectively, adjusting for categorical periods and stratum effects. The GEE analysis assumes marginal Bernoulli distributions for any antibiotic use and employs bias-corrected standard errors to adjust for the moderately small number of clusters (villages).

[3]The correlation parameters for a nested exchangeable working correlation structure are estimated with MAEE and reported for the logistic model only (see **S1 Appendix** and **S4 Table**).

GEE, generalized estimating equation; iCCM, Integrated Community Care Management; MAEE, matrix-adjusted estimating equation; mRDT, malaria rapid diagnostic test; SCJA, Sick Child Job Aid.

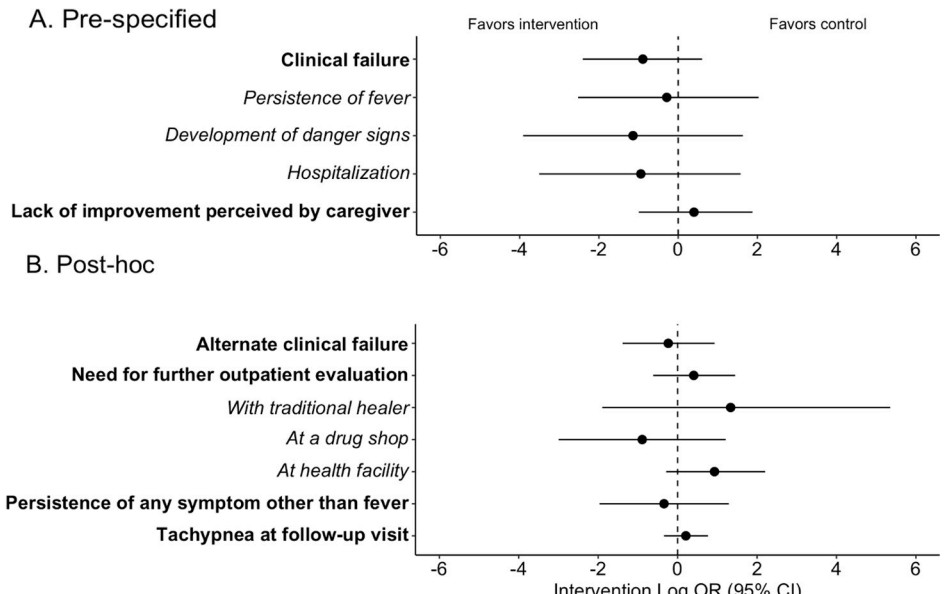

**Fig 4. Forest plot of secondary outcomes.** (A) Prespecified secondary outcomes. (B) Post hoc secondary outcomes. Odds ratios presented on the log scale with point estimates shown by circles and 95% confidence intervals shown by error bars. (Non-transformed ORs with 95% CIs and *p*-values are detailed in **S5 Table**.) Dashed line indicates null value. Individual components of composite outcomes indicated with italic text below the corresponding composite outcome.

result being 40 to 80 mg/l (58.6% (371/632)). Only 11.6% of participants had CRP levels of >80 mg/l. When stratifying by mRDT result, a smaller proportion of children with a CRP level of <10 mg/l was mRDT-positive (4.1%, 13/317) than mRDT-negative (10.2%, 32/313).

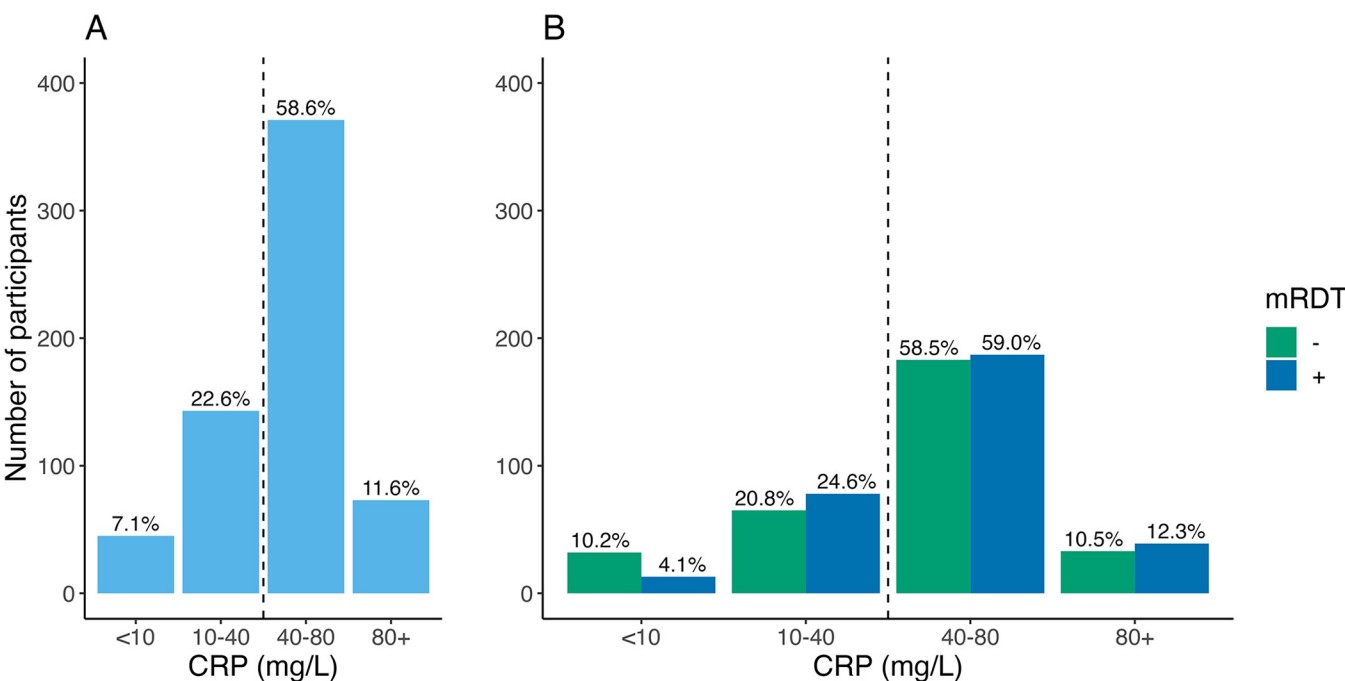

**Fig 5. Distribution of CRP measurements from children enrolled during the intervention periods for (A) all participants and (B) by mRDT result.** CRP, C-reactive protein; mRDT, malaria rapid diagnostic test.

**Table 4. Algorithm adherence for intervention (CRP, top) and control (tachypnea and cough, bottom) groups[1].**

| CRP (mg/l) | Antibiotic use | | Total |
| --- | --- | --- | --- |
| | No | Yes | |
| <40<br>n (%) | 185 (98.4%) | 3 (1.6%) | 188 |
| 40+<br>n (%) | 0 (0%) | 444 (100%) | 444 |
| Total | 185 | 447 | 632 |
| Tachypnea[2]<br>and cough | Antibiotic use | | Total |
| | No<br>n (%) | Yes<br>n (%) | |
| No<br>n (%) | 38 (49.4%) | 39 (50.6%) | 77 |
| Yes<br>n (%) | 10 (2.0%) | 497 (98.8%) | 507 |
| Total | 48 | 536 | 584 |

[1]CRP measurements were missing for one participant and RR was missing for 3 children.

[2]Tachypnea was defined as RR $>= 50$ for children 2–11 months and RR $>= 40$ for children 12 months and older. CRP, C-reactive protein; RR, respiratory rate.

## Adherence to algorithm

Antibiotic prescriptions by CRP test result are shown in **Table 4**. Of those with CRP <40 mg/l, 1.6% (3/631) received antibiotics despite the STAR SCJA advising against it. A CHW noted that for one of these participants, the caregiver insisted on an antibiotic prescription.

Per the iCCM SCJA, elevated respiratory rate for age determines which children with cough should be given antibiotics. Among participants in the control condition without tachypnea and cough, 50.6% (39/77) received antibiotics. There were also 10 children (2%) who were tachypneic and had cough who did not receive antibiotics, 6 of whom were mRDT-positive.

## Associated costs

Purchasing the CRP tests directly from the manufacturer, the cost per test was 1.53 EUR (approximately 1.70 USD). Assuming a 25% decrease in antibiotic use as observed in this study, the number needed to test to avoid one course of antibiotics would be 4 with an associated cost of USD 6.80. A course of amoxicillin treatment for a child aged 1 to 5, if purchased from a local drug shop, ranges from 2,000 to 5,000 UGX (approximately 0.50 to 1.30 USD) depending on country of manufacture.

## Discussion

This article reports results from a stepped wedge cluster randomized trial of an adaptation of the iCCM algorithm, the STAR SCJA, that employs point-of-care CRP measurement by CHW to guide antibiotic treatment decisions for children presenting with febrile ARI in rural western Uganda. As compared to the standard iCCM algorithm, the use of the STAR SCJA was associated with a significant decrease in antibiotic use without a statistically significant concomitant increase in adverse outcomes. Importantly, this impact was achieved by non-medically trained, volunteer health workers, who are often the first providers from whom children

seek care in this region. To our knowledge, this is the first study to demonstrate that CHW can implement and interpret point-of-care CRP testing as part of a broader algorithm.

The magnitude of the impact of CRP measurement on antibiotic use was similar to what has been observed in primary care clinics in Vietnam [22,27], but less than what was seen in neighboring Tanzania [31]. The reasons for this are likely several fold. We intentionally chose a lower CRP cut-off for antibiotic prescription (40 mg/l) than what was used in the Tanzanian study (80 mg/l) to favor sensitivity for bacterial infection over specificity as the CHW are not formally medically trained. This choice of cut-off was also supported by work demonstrating a CRP cut-off of 44.1 mg/l as having a negative predictive value of 92% for radiologic end-point pneumonia [23], and a trial conducted in Myanmar and Thailand in which a cut-off of 40 mg/l for antibiotic prescription in febrile illness was safe [33]. In our study, only 12% of participants had CRP levels of >80 mg/l. Therefore, assuming complete adherence by providers to the algorithm, had that higher cut-off been used, we would have observed similar decreases in antibiotic use as the Tanzanian study. In addition, in the Tanzanian study, the CRP test was included as part of a package of multiple rapid diagnostics in the context of an electronic clinical decision support algorithm for all pediatric febrile illness that, for evaluation of children with respiratory illness, also incorporated respiratory rate thresholds that accounted for age and temperature. The incorporation of multiple stewardship interventions may have led to additional decreases in antibiotic use. Finally, a higher proportion of children were mRDT-positive in this study than anticipated based on our previous research in the region and as compared to other similar studies [31,51]. As malaria infection itself is associated with increased CRP levels [52], it may have contributed to the high proportion of children with CRP levels 40 to 80 mg/l (approximately 60%) who received antibiotics per protocol (see **Fig 5**).

Interestingly, a study in Myanmar and Thailand observed a smaller absolute difference in antibiotic prescribing (risk difference of 5%) using the same CRP cut-off as our study. This disparity may be due to differences in study population (it included both children and adults presenting with any febrile illness to primary care clinics) and lower baseline frequency of antibiotic prescribing (39% in the control group). In addition, in comparison to this and other pragmatic evaluations of CRP use, we observed very high uptake of CRP use and adherence to the study algorithm in our study [27,30,32]. Our research group has been conducting studies with the CHW in this region for 10 years, and overall, community perception of health-related interventions is positive. The local CHW have previously demonstrated high adherence to routine iCCM algorithms [18,37], and anecdotally, the CHWs reported confidence in the CRP test and appreciation of having an additional tool to inform their treatment decisions. We conducted a qualitative sub-study of CHW who participated in the study to examine this further, the results of which will be published separately.

Importantly, the decrease in antibiotic use that we noted in the intervention condition was not associated with an increase in adverse clinical outcomes, the rates of which were low in both groups. Although the study was not designed or powered to assess non-inferiority of the STAR SCJA for clinical failure (a rare outcome) or the other safety outcomes, including hospitalization, our results are consistent with numerous other studies that have demonstrated the safety of CRP testing for antibiotic stewardship [22,27,30–32]. Future studies of CRP use by CHW should be powered to detect differences in clinical outcomes. Mortality rates in children with pneumonia in resource-constrained settings remain high relative to other parts of the world despite widespread use of highly sensitive iCCM algorithms that lead to overprescription of antibiotics [53,54]. Therefore, improving, or at the very least maintaining, the identification of children who are higher risk of poor outcomes is a crucial goal when seeking to optimize clinical algorithms for pneumonia. When CRP measurement was included as part of

an electronic clinical decision support algorithm with multiple point-of-care tests employed at health facilities, the algorithm was noninferior regarding clinical outcomes as compared to usual care [32]. It will be important to ensure this holds true outside of health facilities with subsequent research. In addition, future work could compare CRP measurement to, or combine it with, other potential risk stratification tools, such as lactate measurement or pulse oximetry [55,56].

When discussing potential implementation of a diagnostic in a resource-constrained setting, cost is an important consideration [57]. As such, we have reported the associated costs of the Actim CRP test in the context of our study, as well as the usual price for a course of amoxicillin in the study region, although it is important to note that this is not a formal cost-effectiveness analysis. Given the high prevalence of ARI episodes in children, addition of even a point-of-care CRP assay requiring minimal infrastructure to perform may be prohibitively expensive on a national level. Several global organizations, such as the Foundation for Innovative New Diagnostics (https://www.finddx.org/) and the Clinton Health Access Initiative (https://www.clintonhealthaccess.org/) are actively working to improve access to diagnostics in LMIC through innovative partnerships between governments, the private sector, and non-profit organizations, in part through decreasing costs. In addition, it is possible that targeting CRP use to children for which risk of bacterial infection is determined to be low or intermediate after initial clinical assessment, as opposed to all comers with ARI, may maximize its cost-effectiveness [30,31,58]. Future work should more rigorously assess cost-effectiveness by considering the potential impact of patient selection, local test manufacturing, and purchase of assays by government at scale. These analyses should also incorporate cost-savings associated with avoidance of unnecessary antibiotic use including the potential decrease in negative clinical outcomes (such as adverse effects associated with antibiotic use) and AMR.

Our study had numerous strengths. By adding the CRP test to routine iCCM care provided by non-professional CHWs, we extend the previous research in health facilities to a novel context—one which represents the first point of healthcare contact for many caregivers with ill children in rural areas of sub-Saharan Africa and a clinical context in which an inexpensive point-of-care test would be most useful. Furthermore, the cluster randomized design likely minimized risk of between group contamination and the Hawthorne effect as was a concern in previous work [33]. Adherence to the STAR SCJA was high, and there was minimal loss to follow-up. Finally, our study used novel statistical methods for population-averaged models in an integrated approach to study design (and power) and analysis of dichotomous outcomes (with generalized estimating equations) in a stepped wedge design with a moderately small number of clusters (15 villages).

The study, however, also had limitations. First, as described in the Methods section and S2 Appendix, the trial was not prospectively registered, which introduces the potential for investigator bias. However, the risk of this bias is thought to be minimal as the study protocol and analysis plan were prespecified and received ethical approval prior to start of enrollment (see **S2 Appendix**), the trial was registered prior to the completion of enrollment, and information regarding all enrolled participants is included in this paper. Second, there is a risk of bias in stepped wedge trials with cross-sectional design when identification and recruitment of participants occurs with knowledge of the trial sequence, as was unavoidable given the intervention we evaluated [34]. This risk was likely mitigated by our broad eligibility criteria. In addition, there is a risk of within cluster contamination; only 3 participants received the control condition despite being enrolled in an intervention period, and our analysis was done on an as-treated basis. Third, as mentioned above, because many of the secondary dichotomous outcomes have low rates of events, our study provided relatively poor precision to compare the rates between control and intervention treatment conditions resulting in wide confidence

intervals. Fourth, there were some data collection forms that were lost prior to data entry. Despite this, our sample size exceeded our enrollment estimates, so we still had sufficient power to detect a difference between groups for our primary outcome (see **S1 Appendix**); therefore, these missing data very likely would not have had a significant impact on the study findings. Fifth, we did not systematically evaluate CHW performance of the CRP test in comparison with an expert medical laboratory technician, so we cannot exclude the possibility of misconduct or misinterpretation of the test. To minimize this risk, CHW underwent extensive and repeated trainings as described in the Methods section. Sixth, due to funding constraints, the pulse oximeters used in the study were not designed for children and may have provided inaccurate readings particularly in the youngest children (<2 years of age). Finally, further studies are needed to assess feasibility and acceptability of the STAR SCJA and CRP test when implemented by community health worker programs elsewhere in Uganda and in other countries to ensure our findings are broadly generalizable.

In conclusion, the addition of point-of-care CRP measurement to routine iCCM care by CHW is feasible and has the potential to reduce unnecessary antibiotic use outside of the formal health sector. These findings support expanded, more equitable access to simple diagnostics to improve antibiotic stewardship in rural, resource-constrained settings where individuals with limited medical training are often the first providers from whom ill children seek care.

## Supporting information

**S1 Checklist. CONSORT SW-CRT Checklist.** Checklist of information to include when reporting a stepped wedge cluster randomized trial (SW-CRT).
(PDF)

**S1 Fig. STAR STUDY SICK CHILD JOB AID FOR INTERVENTION PERIODS.** Images were hand drawn by author Baguma Emmanuel.
(PDF)

**S2 Fig. Number of children enrolled per sequence and period.** The study began enrollment on November 1, 2021 and ended enrollment on May 12, 2022 with each period lasting approximately 1 month. Yellow blocks represent control periods; blue blocks are intervention periods. The number shown within the blocks is the total number of participants enrolled for each sequence and period. The number of CHW in each sequence is shown in parentheses next to the name of the villages.
(DOCX)

**S3 Fig. Study enrollment by participating community health worker (CHW).** The number of children enrolled by each CHW grouped by village and treatment sequence. CHW were randomly assigned numbers 1 through 5 for this figure.
(DOCX)

**S4 Fig. Percentage of children who were given or prescribed antibiotics by treatment sequence and period.** Yellow blocks represent control periods; blue blocks are intervention periods.
(DOCX)

**S5 Fig. Percentage of children who were given or prescribed antibiotics by village.** Yellow blocks represent control periods; blue blocks are intervention periods.
(DOCX)

**S1 Table. Scheme for stratification of participating villages.** One village was randomly selected from each stratum for each sequence. Proportions of antibiotic use shown are unadjusted.
(DOCX)

**S2 Table. Demographic and occupational characteristics of the CHW who participated in the STAR Study.** Continuous variables are shown as medians with interquartile ranges, whereas categorical variables are shown as $n$ (%). $n = 65$ except where specified.
(DOCX)

**S3 Table. Percentage of participants who were given or prescribed antimalarial treatment by mRDT result and treatment condition.**
(DOCX)

**S4 Table. Generalized estimating equations (GEEs) estimates and standard errors for logistic regression of antibiotic use.**
(DOCX)

**S5 Table. Frequency of secondary outcomes and estimated odds ratios for association with the intervention.**
(DOCX)

**S1 Appendix. Supplementary Methods.**
(DOCX)

**S2 Appendix. Supplemental Regulatory Materials.**
(PDF)

## Acknowledgments

We sincerely thank the Village Health Teams for their enthusiastic implementation of this study. We are also greatly appreciative of all the children and caregivers who participated in the study and the local community leaders for their support of this work. Finally, thanks to Hinz Printery in Kasese for assistance with graphic design of the STAR SCJA.

The content of this paper is solely the responsibility of the authors and does not necessarily represent the official views of the NIH.

## Author Contributions

**Conceptualization:** Emily J. Ciccone, John S. Preisser, Jonathan J. Juliano, Edgar M. Mulogo, Ross M. Boyce.

**Data curation:** Emily J. Ciccone, Di Hu, John S. Preisser, Caitlin A. Cassidy.

**Formal analysis:** Di Hu, John S. Preisser, Caitlin A. Cassidy.

**Funding acquisition:** Emily J. Ciccone.

**Investigation:** Emily J. Ciccone, Lydiah Kabugho, Baguma Emmanuel, Georget Kibaba, Fred Mwebembezi, Edgar M. Mulogo.

**Methodology:** Emily J. Ciccone, John S. Preisser, Jonathan J. Juliano, Ross M. Boyce.

**Project administration:** Emily J. Ciccone, Lydiah Kabugho, Baguma Emmanuel.

**Resources:** Emily J. Ciccone, Lydiah Kabugho, Baguma Emmanuel, Georget Kibaba, Fred Mwebembezi.

**Supervision:** Emily J. Ciccone, Baguma Emmanuel, Jonathan J. Juliano, Edgar M. Mulogo, Ross M. Boyce.

**Validation:** Emily J. Ciccone.

**Visualization:** Emily J. Ciccone, Di Hu, John S. Preisser, Caitlin A. Cassidy.

**Writing – original draft:** Emily J. Ciccone, Di Hu, John S. Preisser, Caitlin A. Cassidy.

**Writing – review & editing:** Emily J. Ciccone, Di Hu, John S. Preisser, Caitlin A. Cassidy, Lydiah Kabugho, Baguma Emmanuel, Georget Kibaba, Fred Mwebembezi, Jonathan J. Juliano, Edgar M. Mulogo, Ross M. Boyce.

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
