## [Editor Report · Decision Letter 0]

20 May 2024

Dear Dr Ciccone, 

Thank you for submitting your manuscript entitled "Point-of-care C-reactive protein measurement by community health workers safely reduces antimicrobial use among children with respiratory illness: results from a stepped wedge cluster-randomized trial in rural Uganda" for consideration by PLOS Medicine.

Your manuscript has now been evaluated by the PLOS Medicine editorial staff as well as by an academic editor with relevant expertise and I am writing to let you know that we would like to send your submission out for external peer review.

As discussed with you via direct email correspondence, the editorial team also has the following request:

1. Please add a paragraph to the Methods section explaining the timeline of registration and recruitment. You may wish to add a summary outline timeline in your Method section and include a more detailed timeline in an appendix. 

2. In the Methods section, please also provide a reason for non-prospective registration. 

3. Please also acknowledge the Method section the possible introduction of bias due to late registration, emphasising the potential for increased investigator bias. Please add this also to your limitations. This is important because when future evidence syntheses are done, it will be clear to all readers that this may be a reason to downgrade the risk of bias assessment. 

3. Please add V2.0 and V3.0 study protocols of your study as SI (this way, the readers can see for themselves how little has changed) AND please also add the IRB approval docs when V2.0 was approved as this provides firm dates.

4. Please reported in accordance with CONSORT. Please complete the CONSORT checklist and ensure that all components of CONSORT are present in the manuscript. When completing the checklist, please use section headers and paragraph numbers, rather than page numbers (as the published article will not have page numbers.)

Please re-submit your manuscript within two working days, i.e. by May 23 2024 11:59PM.

Feel free to email me at kjanin@plos.org if you have any queries relating to your submission.

Kind regards,

Katrien G. Janin, PhD

Senior Editor

PLOS Medicine

---

## [Decision Letter · Decision Letter 1]

4 Jun 2024

Dear Dr. Ciccone,

Thank you very much for submitting your manuscript "Point-of-care C-reactive protein measurement by community health workers safely reduces antimicrobial use among children with respiratory illness: results from a stepped wedge cluster-randomized trial in rural Uganda" (PMEDICINE-D-24-01411R1) for consideration at PLOS Medicine. 

As you will see, the reviewers were very positive about the paper and the importance of these follow-up data, but they raised a number of questions about specific study details, methodology, and presentation. After discussing the paper with the editorial team and an academic editor with relevant expertise, I’m pleased to invite you to revise the paper in response to the reviewers’ comments. We plan to send the revised paper to all of the original reviewers, and of course we cannot provide any guarantees at this stage regarding publication.

When you upload your revision, please include a point-by-point response that addresses all of the reviewer and editorial points, indicating the changes made in the manuscript and either an excerpt of the revised text or the location (eg: page and line number) where each change can be found. Please submit a clean version of the paper as the main article file and a version with changes marked should as a marked-up manuscript. Please also check the guidelines for revised papers at http://journals.plos.org/plosmedicine/s/revising-your-manuscript for any that apply to your paper.

We also request that you upload any figures associated with your paper as individual TIF or EPS files with 300dpi resolution at resubmission; please read our figure guidelines for more information on our requirements: http://journals.plos.org/plosmedicine/s/figures. While revising your submission, please upload your figure files to the PACE digital diagnostic tool, https://pacev2.apexcovantage.com/. PACE helps ensure that figures meet PLOS requirements. To use PACE, you must first register as a user. Then, login and navigate to the UPLOAD tab, where you will find detailed instructions on how to use the tool. If you encounter any issues or have any questions when using PACE, please email us at PLOSMedicine@plos.org.

We look forward to receiving your revised manuscript. 

Sincerely,

Katrien Janin, PhD

PLOS Medicine

plosmedicine.org

Comments from the Academic Editor:

1) The authors have not provided a rational for why the step-wedge design was needed. My feeling here is that a parallel group cluster randomised trial would have been a better choice, and would be be at lower risk of bias. One justification for step-wedge trials is that the intervention is going to be introduced into routine practice, and so this design allows evaluation during implementation. If the authors want to make this argument, it should be justified by strong evidence showing that point of care CRP is now widely implemented and maintained by the Ministry of Health in this setting. Please justify the selection of step-wedge design in terms of scientific rationale to address their hypothesis (rather than logistics).

2) Statistical analysis: mostly rigorously done. However, given the stratified randomisation schedule, the authors should have probably adjusted regression model estimates of effect for stratification variables - not doing so usually leads to inflated effect sizes (https://trialsjournal.biomedcentral.com/articles/10.1186/s13063-020-04850-w). This point was also raised by the statistical reviewer.

3) Whilst the author's interpretation of the primary outcome is probably justified, the secondary outcomes are very imprecisely estimated due to small numbers of events. I am not convinced that this trial provides high quality evidence that the intervention did not worse adverse outcomes - it is really very difficult to say one way or the other here. (e.g. main conclusion statement is probably overstated: "“As compared to the standard iCCM algorithm, the use of the STAR SCJA was associated with a significant decrease in antibiotic use without a concomitant increase in adverse outcomes.”

4) Manuscript needs to be reported in accordance with CONSORT statement for stepped wedge cluster-randomized trial (also see point of the statistical reviewer)

6) Expand limitations. Limitations section is a little brief, and perhaps overly optimistic. Would like greater reflection on risk of bias.

1) Please include a short summary of protocol approvals and why the registration was late in the method section. We appreciate you have added a short explanation in the manuscript (Methods), but please expand a bit so that the reader has a short summary of the events without the need to consult the SI. Please keep the full details as SI. Please also acknowledge possible investigator bias this may have introduced. 

2) Reporting guidance

Please use the 'Reporting of stepped wedge cluster randomised trials' CONSORT checklist (apologies for not having made that clear in my previous request) 

3) Data Availability 

PLOS Medicine requires that the de-identified data underlying the specific results in a published article be made available, without restrictions on access, in a public repository or as Supporting Information at the time of article publication, provided it is legal and ethical to do so. Please see the policy at 

http://journals.plos.org/plosmedicine/s/data-availability

and FAQs at 

http://journals.plos.org/plosmedicine/s/data-availability#loc-faqs-for-data-policy

The Data Availability Statement (DAS) requires revision. For each data source used in your study: 

Statistical reporting 

Were relevant, please quantify the main results with 95% CIs and p values. 

When reporting p values please report as <0.001 and where higher as p=0.002, for example. When reporting 95% CIs please separate upper and lower bounds with commas instead of hyphens as the latter can be confused with reporting of negative values. Please include any important dependent variables that are adjusted for in the analyses. 

Abstract layout 

Please ensure your report your abstract according to CONSORT for abstracts, following the PLOS Medicine abstract structure (Background, Methods and Findings, Conclusions) https://www.equator-network.org/reporting-guidelines/consort-abstracts/

Author summary 

At this stage, we ask that you include a short, non-technical Author Summary of your research to make findings accessible to a wide audience that includes both scientists and non-scientists. The authors summary should consist of 2-3 succinct bullet points under each of the following headings: 

• Why Was This Study Done? Authors should reflect on what was known about the topic before the research was published and why the research was needed. 

• What Did the Researchers Do and Find? Authors should briefly describe the study design that was used and the study’s major findings. Do include the headline numbers from the study, such as the sample size and key findings. 

• What Do These Findings Mean? Authors should reflect on the new knowledge generated by the research and the implications for practice, research, policy, or public health. Authors should also consider how the interpretation of the study’s findings may be affected by the study limitations. In the final bullet point of ‘What Do These Findings Mean?’, please describe the main limitations of the study in non-technical language. 

The Author Summary should immediately follow the Abstract in your revised manuscript. This text is subject to editorial change and should be distinct from the scientific abstract. Please see our author guidelines for more information: https://journals.plos.org/plosmedicine/s/revising-your-manuscript#loc-author-summary

Introduction layout 

Please address past research and explain the need for and potential importance of your study. Indicate whether your study is novel and how you determined that. If there has been a systematic review of the evidence related to your study (or you have conducted one), please refer to and reference that review and indicate whether it supports the need for your study. 

Discussion layout 

Please present and organize the Discussion as follows: a short, clear summary of the article's findings; what the study adds to existing research and where and why the results may differ from previous research; strengths and limitations of the study; implications and next steps for research, clinical practice, and/or public policy; one-paragraph conclusion. 

Supplementary materials 

Please note that supplementary materials are not checked and will be posted as supplied by the authors. Therefore, please double check. Please cite your Supporting Information as outlined here: https://journals.plos.org/plosmedicine/s/supporting-information - Please note you may use almost any description as the item name of your supporting information as long as it contains an "S" and number. For example, “S1 Appendix” and “S2 Appendix,” “S1 Table” and “S2 Table. Please ensure each supplementary material has a call out (link) from your main manuscript. 

Comments from the reviewers:

Reviewer #1: This manuscript presents the findings of a stepped wedge cluster randomised trial in 15 villages in Uganda, assessing the effect of a clinical algorithm to the standard approach to guide antibiotic treatment decisions in children with fever and cough. My comments focus on the statistical aspects of this study. The statistical analyses appear to be mostly appropriate (in particular, the use of bias-corrected standard errors to account for a small number of clusters was good to see), as does the interpretation of results. However, there are several issues that require clarification and potentially re-analysis.

1. In Figure 2, please provide a separate row for each village (that is, replace Figure 2 with Figure S2). Please also indicate the length of each period in days in the Period number row. 

2. In Figure 4, why is log OR shown instead of OR? Please display OR if possible.

3. In the Statistical Methods section, it is stated that "Similar models for prevalence differences are also reported…" It seems that these models are more fully described in the footnote of Table 3. Why were terms for period and the stratification factor not included in this model? Please provide results where period and the stratification factor have been included in the model. Why was an independence working correlation structure assumed for this outcome? Provide repeat with a nested exchangeable correlation structure and provide estimates of within-period and between-period correlations.

4. Randomisation into sequences was stratified - were terms for strata included in the outcome regression models? It seems from Table S2 that the stratification factor was not included in the model. Provide estimates where these terms are included.

5. The power calculation assumed a "zero slope temporal trend" - does this mean that a term for period was not included in the power calculation?

6. In Table 1, are the Village stratum labels (small, medium, large) a proxy for altitude and distance from Bugoye? If so, please include the full labels. If not, please also include altitude and distance from Bugoye in Table 1.

7. Include estimates of the within-period and between-period correlations in Table 3.

8. Instead of using the 2010 CONSORT checklist, please use the CONSORT checklist for stepped wedge trials (https://doi.org/10.1136/bmj.k1614). Please ensure that the CONSORT stepped wedge paper is referenced and note the inclusion of the CONSORT checklist in the Supplementary materials in the manuscript.

9. Several post-hoc power calculations have been performed here - however, these have been shown to have no value (e.g. doi: 10.1007/s12144-018-0018-1; there are other references). The provided post-hoc power calculations should be removed entirely.

10. Provide a version of Figure S4 with a separate row for each village, and include n/N along with %.

Reviewer #2: More specific targeting of antimicrobials is an important research area. A ~20% reduction in antibiotic use for ARI in the community is a valuable result. I have relatively few comments.

The step wedge design is a reasonable choice for this kind of intervention. However, seasonality is an issue in step wedge studies as malaria and the predominant cause of ARI in children, RSV, vary in incidence through the year. This potentially affects signs like headache, fast breathing, hypoxia, and tachycardia (as per Table 2). Malaria seems not to have been an issue for antibiotic use - ideally this should have been tested as an interaction/effect modifier. However it is conceivable that CHWs may have altered antimalarial usage. Was there any difference in prescription of antimalarials (by mRDT result)? Potential variation in viral aetiology should be discussed.

Power for the safety outcomes (e.g. hospitalisation) is very low, this should also be mentioned in the discussion. This could be monitored during a larger roll out.

The inclusion of assay cost data is helpful. Given that symptoms of ARI are very common, this may represent a prohibitively high cost to a national programme. Discussion of whether this cost can be reduced either by alternative manufacturing or by using clinical features to select a group with a higher pre-test probability would be useful.

Reviewer #3: Manuscript Number: PMEDICINE-D-24-01411R1

Manuscript title: Point-of-care C-reactive protein measurement by community health workers safely reduces antimicrobial use among children with respiratory illness: results from a stepped wedge cluster-randomized trial in rural Uganda

This is a great manuscript presenting the results of a well-designed and executed clinical trial of the added benefit of introducing a PoC CRP measurement in the decision of prescribing (by CHW) antibiotics to children with acute respiratory infections. It is now well known that clinicians in resource constrained settings, in the absence of supportive diagnostic tools, tend to overprescribe antibiotics guided buy highly sensitive clinical algorithms that prime the use of therapeutics in the context of clinical syndromes of potentially life-threatening infections. This is particularly problematic in the field of pneumonia, where it has previously also been observed that even in the presence of moderate severity, antibiotics do not necessarily improve outcomes in comparison to placebo (possibly due to the fact that many of the moderately severe infections are not responsive to antibiotics, as they tend to be originally viral). Thus, this trial provides important data that could influence policy making in terms of the added value of adding a PoC CRP evaluation at the patient's side to guide management decisions. The results are also important as they suggest the generalized use of such a PoC device could substantially decrease the use of antibiotics WITHOUT WORSENING OUTCOMES, with all the added benefit that ensues, particularly in terms of the growing threat of antimicrobial resistance. I believe the manuscript is well written, clear, and that provides a good summary of the status quo and the context on which the results of the trial could be applied. I have only some minor concerns that I would like to provide, with the hope that the manuscript is strengthened.

* In the differential diagnosis of the clinical syndrome of pneumonia, one needs to also consider malaria in areas where this disease is endemic. It has been shown also that CRP is often similarly elevated in response to acute malaria that in response to bacterial infections. Did authors screen for malaria in the setting where the trial was conducted (rural Uganda, where presumably there is lots of malaria), and where they able to analyse how many mixed infections there were, and how well did the CRP evaluation perform in those cases where malaria infections where confirmed? If this was not done, perhaps this should be acknowledged in the discussion. In this context, and given what CHW are trained to do, where all cases suspected of pneumonia which ALSO had a fever tested for malaria?

* Methods: What kind of amoxicillin was used? Did the trial use conventional or dispersible?

* Method: please specify whether pulsioximetry was used in every recruited patient, and if so whether there were differences in outcomes and performance of CRP according to the presence or absence of hypoxemia. I guess it wasn't used, but then it should be specifically mentioned so that the reader understands this is not routine care by CHWs

* I believe iCCM protocols regarding pneumonia, danger signs, and management recommendations should be more explicitly explained. Perhaps this can be presented as a supplementary table or panel

* In the methods section please define: what is the HIV prevalence rate estimated in the area. 

* What are the routine vaccines against pneumonia currently in use in the area. Where there changes or additions of new vaccines or different ones throughout the development of the trial?

* Inclusion and exclusion criteria: "if the previous illness had completely resolved, and they were re-presenting for a new episode of ARI". Was there a lag period considered for this re-inclusion? In other words, if a child presented with pneumonia, and somehow resolution of the symptoms, was there a minimum number of days that needed to pass before considering the child again eligible?

* I believe what the pulsioximeter defines is hypoxemia rather than hypoxia. Please verify

* Was weight measured? If yes, wouldn't it be better to define malnutrition to use weight rather than MUAC?

* I believe a brief description of the CHWs may be useful to understand whether there could be important differences among them in terms of training and expertise, that could somehow have confounded the results

* Regarding CRP thresholds and values, as obtained at the PoC. What were the recommendations (and training) given to the CHW in terms of how should they interpret CRP results?

* As one of the secondary endpoints was death, you should state whether there were any deaths in the trial and in each of the groups

* I believe the small paragraph on costs and estimated costs averted is useful, but either you should include a bit more detailed info or tone down the conclusions, given that the study was not really designed to assess cost effectiveness

* The manuscript does not discuss one of the hot topics of risk stratification and triage issues more relevant to work in settings with scarcity of resources, which is the idea of prognosis vs. diagnosis. Given that your intervention (PoC CRP test) is essentially a guide to ascertain whether the underlying cause of pneumonia is of likely bacterial or non bacterial origin), your endpoint is rightly designed as use of antibiotics. It could also very well be that CRP tells you, in advance, which children are more likely to go well (or unwell, and have an adverse outcome) with prognosis becoming more important. I would personally appreciate if the authors could reflect a bit on these implications, and about the different risk stratification approaches to improve pneumonia outcomes.

* Table 1: besides guardian's occupation, do you have info on guardian's level of education? This may be more relevant in relation to outcomes than occupation

* Regarding the analysis stratified by malaria RDT, do you have also data on those who received (or did not) an antimalarial? Could you also stratify by reception (or not) of antimalarials?

* Figure 5 could benefit from adding a vertical line in the value considered threshold for recommendation of antibiotic use

[LINK]

---

## [Decision Letter · Decision Letter 2]

17 Jul 2024

Dear Dr. Ciccone,

Thank you very much for re-submitting your manuscript "Point-of-care C-reactive protein measurement by community health workers safely reduces antimicrobial use among children with respiratory illness: results from a stepped wedge cluster randomized trial in rural Uganda" (PMEDICINE-D-24-01411R2) for review by PLOS Medicine.

I have discussed the paper with my colleagues and the academic editor and it was also seen again by the reviewers. I am pleased to say that provided the remaining editorial and production issues are dealt with we are planning to accept the paper for publication in the journal.

[LINK]

If you have any questions in the meantime, please contact me (kjanin@plos.org) or the journal staff on plosmedicine@plos.org.  

We look forward to receiving the revised manuscript by Jul 24 2024 11:59PM.   

Sincerely,

Katrien Janin, PhD

Senior Editor 

PLOS Medicine

plosmedicine.org

Requests from Editors:

Thank you for your detailed response to the editors' and reviewers' comments. I have discussed the paper with my colleagues and the academic editor, and it has also been seen again by the original reviewers. The changes made to the paper were satisfactory to the reviewers and editors alike.

We only have a few minor comments for you to address.

1. Regarding the choice of CRP>40, we appreciate there is some discussion of this in the Discussions section already, but a sentence on this in the Background section would be helpful for the readers.

2. Consent: Please provide additional details regarding participant consent. Your study included minors, and you state that you obtained consent from parents or guardians. In the ethics statement in the Methods and online submission information, please ensure that you have specified what type you obtained (for instance, written or verbal, and if verbal, how it was documented and witnessed).

3. In Table 1. Demographics of the study population, one of the occupations listed is 'peasant'. I wondered if it would be better to label this to 'agricultural laborer' or something alike, given the negative connotations associated with the word 'peasant'.

4. References: For in-text reference, citations are placed within square parentheses [ ] and should precede punctuation. Please amend. 

PLOS uses the numbered citation (citation-sequence) method and first six authors, et al. 

Please ensure that journal name abbreviations match those found in the National Center for Biotechnology Information (NCBI) databases (http://www.ncbi.nlm.nih.gov/nlmcatalog/journals), and are appropriately formatted and capitalised. 

For e.g reference 12, 37, 38, 39, etc : Where website addresses are cited, please specify the date of access (e.g. add [accessed: 16/09/2023] - instead of [cited]). Please amend.

Please see https://journals.plos.org/plosmedicine/s/submission-guidelines#loc-references for further details on reference formatting.  

5. ACKNOWLEDGMENTS/ DECLARATIONS

Please remove all statements apart from acknowledgements, author contributions and abbreviations from the end of the main manuscript and include these only in the relevant parts of the manuscript submission form. Funding, competing interest, and data availability will be compiled as metadata.

General comment:

This is a kind reminder that supplementary materials are not checked and will be posted as supplied by the authors. Therefore, please double check. Please cite your Supporting Information as outlined here: https://journals.plos.org/plosmedicine/s/supporting-information - Please note you may use almost any description as the item name of your supporting information as long as it contains an "S" and number. For example, “S1 Appendix” and “S2 Appendix,” “S1 Table” and “S2 Table. 

Please also ensure each supplementary material has a call out (link) from your main manuscript.

We have given you a week to resubmit, but given the minor nature of the above request, feel free to submit your revised version sooner.

Comments from Reviewers:

Reviewer #1: I thank the authors for their responses to my comments on the previous version of this manuscript. I have no further comments. 

Reviewer #2: My comments have been fully addressed.

Reviewer #3: Authors have adequately responded (in my opinion) all reviewer's comments. The manuscript is in good shape now.

[LINK]

---

## [Editor Report · Decision Letter 3]

24 Jul 2024

Dear Dr Ciccone, 

On behalf of my colleagues and the Academic Editor, I am pleased to inform you that we have agreed to publish your manuscript "Point-of-care C-reactive protein measurement by community health workers safely reduces antimicrobial use among children with respiratory illness: results from a stepped wedge cluster randomized trial in rural Uganda" (PMEDICINE-D-24-01411R3) in PLOS Medicine.

Please note: I only have one small additional editorial request for you (and with my apologies for not having picked up on this earlier). Please change your title to "Point-of-care C-reactive protein measurement by community health workers safely reduces antimicrobial use among children with respiratory illness in rural Uganda: A stepped wedge cluster randomized trial". The reason for this request is that the second part of the title is reserved for the study descriptor as per PLOS title style guide. 

PRESS

Thank you again for submitting to PLOS Medicine. We look forward to publishing your paper and we hope we may work together again in the future.

Sincerely, 

Katrien G. Janin, PhD 

Senior Editor 

PLOS Medicine